# Influence of Fixation Methods on Prosthetic Joint Infection Following Primary Total Knee Replacement: Meta-Analysis of Observational Cohort and Randomised Intervention Studies

**DOI:** 10.3390/jcm8060828

**Published:** 2019-06-11

**Authors:** Setor K. Kunutsor, Vikki Wylde, Michael R. Whitehouse, Andrew D. Beswick, Erik Lenguerrand, Ashley W. Blom

**Affiliations:** 1National Institute for Health Research Bristol Biomedical Research Centre, University Hospitals Bristol NHS Foundation Trust and University of Bristol, Bristol BS8 2BN, UK; V.Wylde@bristol.ac.uk (V.W.); Michael.Whitehouse@bristol.ac.uk (M.R.W.); Ashley.Blom@bristol.ac.uk (A.W.B.); 2Translational Health Sciences, Bristol Medical School, Musculoskeletal Research Unit, University of Bristol, Learning & Research Building (Level 1), Southmead Hospital, Bristol BS10 5NB, UK; Andy.Beswick@bristol.ac.uk (A.D.B.); Erik.Lenguerrand@bristol.ac.uk (E.L.)

**Keywords:** fixation, cemented, uncemented, hybrid, antibiotic-loaded cement, prosthetic joint infection, total knee replacement, meta-analysis

## Abstract

The type of fixation used in primary total knee replacement (TKR) may influence the risk of prosthetic joint infection (PJI). We conducted a systematic review and meta-analysis to assess published evidence linking type of fixation (cemented, uncemented, or hybrid) with the risk of PJI following primary TKR. Randomised controlled trials (RCTs) and observational cohort studies comparing fixation methods and reporting PJI incidence following primary TKR were identified in MEDLINE, Embase, Web of Science, and Cochrane Library up until November 2018. Summary measures were relative risks (RR) with 95% confidence intervals (CIs). We identified 32 eligible articles (24 observational studies and 8 RCTs) involving 1,161,292 TKRs. In pooled analysis of observational studies, uncemented fixation was associated with a decreased overall PJI risk when compared with cemented fixation at 0.76 (0.64–0.89). Comparing antibiotic-loaded cemented fixation with plain cement, there was no significant difference in overall PJI risk at 0.95 (0.69–1.31), but PJI risk was increased in the first 6-month postoperative period to 1.65 (1.12–2.43). Limited data from RCTs showed no differences in PJI risk among the fixation types. Observational evidence suggests uncemented fixation may be associated with lower PJI risk in primary TKR when compared with cemented fixation. In the early postoperative period, antibiotic-loaded cemented fixation may be associated with increased PJI risk when compared with plain cement. This may either reflect appropriate selection of higher risk patients for the development of PJI to cemented and antibiotic-loaded cement or may reflect a lower PJI risk in uncemented TKR due to factors such as shorter operative time.

## 1. Introduction

Total knee replacement (TKR) is one of the most common elective surgical procedures performed worldwide. In 2017 alone, 102,777 TKRs were performed in England, Wales, Northern Ireland, and the Isle of Man, as recorded in the National Joint Registry (NJR) [1]. In a primary TKR, the knee implants (femoral and tibial components) may be secured to the bone with (cemented) or without (uncemented) bone cement (i.e., type of fixation). The TKR construct fixation is referred to as cemented if the femoral and tibial implants are bonded to the bone using cement; in uncemented fixation, the femoral and tibial implants use press-fit into the bone for initial stability and then bone ingrowth into coatings on the structure of the implant without cement; in hybrid fixation, there is a mixture of fixations, with one of the implants being cemented and one being uncemented.

Although TKR is often a successful intervention for alleviating pain and improving function in joint disease, such as osteoarthritis [2], some patients experience complications, such as aseptic loosening, prosthetic joint infection (PJI), chronic pain, instability, malalignment, and wear [1]. PJI is a rare but dreaded complication of TKR; it affects between 0.4–1.5% of primary TKRs [3]. PJI and its management has devastating effects on patients [4] and it is associated with significant morbidity [5,6,7], as well as with high healthcare costs [8,9]. With increasing life expectancy and number of people who will be affected by osteoarthritis, there will be a rise in the numbers of TKRs and the number of patients affected by PJI is also expected to increase in a proportionate manner [10,11]. In England and Wales, over a thousand revision operations are performed annually due to PJI of the knee [12]. 

Patient-, surgery-, and health-system-related factors influence the risk of developing PJI following a knee replacement [13,14]. Whether surgery-related factors such as fixation methods influence the risk of developing a PJI following total joint replacement has been a controversial issue, as the evidence has been inconsistent. In a recent review, our group showed that compared with other fixation methods, uncemented and antibiotic-loaded cemented fixations carry the lowest risk for PJI following total hip replacement (THR) [15]. Data on whether fixation methods affect PJI rates differentially following TKR remains uncertain, as the literature is conflicting. In this context, we aimed to evaluate the body of evidence linking cemented, uncemented, and hybrid fixation methods with the risk of PJI following primary TKR, using a systematic review and meta-analysis of both observational and randomised trial evidence. Our specific objectives were: (i) to compare the nature and magnitude of potential associations of different fixation methods with risk of PJI; and (ii) to assess if the associations varied by study and individual level characteristics.

## 2. Experimental Section

### 2.1. Data sources and Search Strategy

This review was conducted according to PRISMA and MOOSE guidelines [16,17] (Appendix A) and was based on a pre-defined protocol, which has been registered in the PROSPERO International prospective register of systematic reviews (CRD42018114592). We searched MEDLINE, Embase, and The Cochrane Library for studies comparing two or more of the following fixation types: cemented, uncemented, and hybrid, and reported PJI outcomes after primary TKR from inception to November 2018. The computer-based searches combined free and MeSH search terms and combinations of keywords related to the target population (e.g., “total knee replacement”, “total knee arthroplasty”, “total joint replacement”), the intervention (e.g., “fixation”, “cemented”, “uncemented”, “cementless”, “hybrid”), and outcome (e.g., “prosthetic joint infection”, “deep infection”, “infection”). The search was limited to human studies with no restrictions on language. The detailed search strategy is reported in Appendix A. All titles and abstracts of studies retrieved from the databases were initially screened to assess their suitability for inclusion. Full-text evaluation of articles potentially meeting eligibility criteria was conducted independently by two authors (S.K.K. and V.W.) for study selection. Any disagreements regarding eligibility of an article were discussed and consensus was reached with involvement of a third author (M.R.W) when necessary. Reference lists of eligible articles and relevant review articles were manually scanned for additional studies not identified by our original search. Citations of key studies were checked in Web of Science.

### 2.2. Eligibility Criteria

Studies were eligible if they were comparative observational cohort designs, case-control designs, or randomised controlled trials (RCTs) that: (i) recruited participants undergoing primary TKR; (ii) compared any two or more of the following fixation types: cemented, uncemented, and hybrid fixation; and (ii) reported PJI outcomes after a period of follow-up following primary TKR. No restrictions were imposed on the follow-up duration. We excluded the following studies: (i) the intervention was based on only revision TKR; (ii) studies of only bilateral TKR or studies reporting paired fixations (e.g., cemented and uncemented TKR in the same patient); (iii) compared fixation methods of only one component (tibial or femoral) and did not provide information on the fixation type of the other component; (iv) and those conducted in selected populations (e.g., patients with diabetes only).

### 2.3. Data Extraction and Quality Assessment

One reviewer (S.K.K.) initially conducted the data extraction using a standardised data collection form. A second reviewer (V.W.) independently checked the extracted data with that in the original articles. Data on the following were extracted: first author’s name, study publication date, country and geographical location of study, study design, baseline year, mean age, duration of follow-up, sample size, intervention and control, number of PJI outcomes, risk estimates (relative risks (RRs), hazard ratios (HRs), or odds ratios (ORs)), and degree of covariate adjustment (univariable or multivariable). We assessed the methodological quality of observational studies using the Newcastle-Ottawa Scale (NOS) [18]. This scale is used for assessing the quality of non-randomised studies and uses a star system that rates the quality of evidence from a score of zero to nine, based on three domains: selection of participants; comparability of study groups; and ascertainment of outcomes of interest. The Cochrane Collaboration’s risk of bias tool was used to assess the quality of RCTs [19].

### 2.4. Data Synthesis and Analysis

The risk ratios, expressed as RRs with 95% confidence intervals (CIs), were used as the summary measures of association across studies. Since PJI is considered a rare outcome, reported HRs and ORs were assumed to approximate the same measure of RR following Cornfield’s rare disease outcome assumption [20]. Fully multivariable adjusted risk estimates were used when reported, otherwise crude RRs were calculated from studies that reported raw counts for intervention and control arms. To minimize the effect of heterogeneity, the inverse variance-weighted method was used to pool RRs using random-effects models. We reported RRs of the associations for the overall duration of follow-up. Sub-analyses were also conducted for specific post-operative periods (e.g., first 3–6 months of follow-up) for studies that reported these data. Heterogeneity across studies was assessed using the Cochrane *χ^2^* statistic and the *I^2^* statistic [21]. We explored sources of heterogeneity and assessed for interactions on the associations by pre-defined study-level characteristics using stratified analyses and univariable meta-regression [22]. For pooled analysis involving 10 or more studies, publication bias was assessed by visually inspecting a funnel plot and applying Egger’s regression symmetry test bias [23]. We also adjusted for the effect of publication bias by the use of the Duval and Tweedie’s nonparametric trim-and-fill method, which imputes hypothetical small missing null or negative studies [24]. All statistical analyses were performed with Stata release 15 (StataCorp, College Station, TX, USA).

## 3. Results

### 3.1. Study Identification and Selection

The literature search strategy, manual scanning of reference lists, and citation check of Web of Science identified 665 potentially relevant articles. After the initial screening of titles and abstracts, 51 articles remained for full text evaluation. Following detailed full text evaluation, 19 articles were excluded because: (i) the outcome was not relevant (*n* = 8); (ii) intervention was not relevant (*n* = 7); (iii) population was not relevant (*n* = 2); and (iv) duplicate studies (*n* = 2). The remaining 32 articles based on 24 unique observational cohort studies and eight RCTs met the inclusion criteria and were included in the review (Figure 1; Table 1; Appendix A).

### 3.2. Study Characteristics and Study Quality

Table 1 provides key characteristics of eligible observational cohort studies and RCTs included in the review. Overall, the 32 studies included 1,161,292 TKRs and 5706 PJI outcomes. The 24 observational cohort studies included 1,157,263 TKRs and 5598 PJI outcomes. Of the 24 studies, 11 were conducted in North America (USA and Canada), eight in Europe (Austria, Croatia, Finland, France, Norway, Spain, and the United Kingdom), three in the Pacific region (Australia and New Zealand), and two in Asia (China and Taiwan). Observational studies were published between 1990 and 2018. The population sources from which these studies were based included arthroplasty and community registries, hospitals, and institutional databases. The mean/median baseline age of participants ranged from 59 to approximately 76 years. PJI outcomes were reported in a variety of ways and included revision for infection, deep infection, and surgical site infection (Table 1). Majority of studies, especially the registry studies, did not provide any detailed information on the diagnoses of PJI. For studies reporting the diagnoses of infection, the definitions varied but were mostly based on criteria developed by Centers for Disease Control Prevention (CDC) [25] and the Musculoskeletal Infection Society (MSIS) [26]. For registry studies, a previously published and related study has indicated that reporting of infection as the cause of revision in registry studies reflects the surgeon’s opinion based on clinical information and findings at surgery [27]. The average overall follow-up for PJI outcomes ranged from one year to approximately 14 years. Some studies also reported early postoperative follow-up results within periods of 30 days to 6 months. The NOS methodological quality of included observational studies ranged from 4–8.

Of the eight RCTs, five were conducted in Europe (France, Spain, and the United Kingdom), two in North America (Canada and the United States), and one in Asia (Korea). These trials were published between 1998 and 2015. Altogether, the RCTs comprised 4029 TKRs and 108 PJI outcomes, with sample sizes ranging from 81 to 2948 TKRs. The average duration of follow-up for PJI outcomes ranged from 2 to 9.4 years. Using the Cochrane risk of bias tool, all trials demonstrated a high risk of bias within 1–5 areas of study quality (random sequence generation, allocation concealment, blinding of participants and personnel, blinding of outcome assessment, and other bias). All trials had a low risk of bias for incomplete outcome data and selective reporting. Four trials had an unclear risk of bias in allocation concealment (Appendix A).

### 3.3. Fixation Types and PJI Risk

Figure 2 reports RRs (95% CIs) for overall PJI, comparing various fixation types for all studies. In observational studies, compared with cemented fixation, uncemented fixation was associated with a lower risk of PJI (8 studies, 892,094 TKRs, and 4118 PJIs) RR 0.76 (95% CI: 0.64–0.89) (Appendix A). There was no significant evidence of heterogeneity between contributing studies (*I*^2^ = 14%; 95% CI: 0–57%; *p* = 0.318). When the largest study, which was based on the NJR [14], was excluded from the analysis, the pooled RR was 1.45 (95% CI: 0.85–2.48). There was no significant difference in PJI risk when hybrid fixation was compared with cemented RR 0.98 (95% CI: 0.80–1.21) or uncemented fixation RR 0.93 (95% CI: 0.13–6.25) (Figure 2; Appendix A). In pooled analysis of 12 observational studies (200,442 TKRs, and 1039 PJIs), there was no significant difference in overall PJI risk when antibiotic–loaded cemented fixation was compared with plain cemented fixation RR 0.95 (95% CI: 0.69–1.31) (Figure 2; Appendix A). There was evidence of significant heterogeneity between contributing studies (*I*^2^ = 71%; 95% CI: 47–84%; *p* < 0.001), which was partly explained by geographical location (*p* for meta-regression < 0.001) and population source (joint registries versus other data sources) (*p* for meta-regression = 0.035) (Figure 3). Antibiotic-loaded fixation was associated with decreased PJI risk in Asian populations, with no difference in risk in other geographical locations. In further analysis limited to PJI outcomes at 6 months of follow-up in studies providing these data (3 studies, [28,29,30] 74,955 TKRs, 147 PJIs), antibiotic-loaded cemented fixation was associated with an increased risk of PJI when compared with plain cemented fixation RR 1.65 (95% CI: 1.12–2.43). There was no evidence of heterogeneity between contributing studies (*I*^2^ = 0%; 95% CI: 0–90%; *p* = 0.561). In pooled analysis of 4 studies (48,961 TKRs, 177 PJIs) restricted to PJI diagnosed at 24 or more months of follow-up, there was no significant difference in PJI risk when antibiotic-loaded cemented fixation was compared with plain cemented fixation RR 0.73 (95% CI: 0.33–1.63).

In RCTs, there was no difference in PJI risk when uncemented fixation was compared with cemented or hybrid fixation and when hybrid fixation was compared with cemented fixation (Figure 2; Appendix A). In one trial based on 2948 TKRs and 85 PJIs, [31], there was no difference in PJI risk when antibiotic-loaded cemented fixation was compared with plain cemented fixation RR 1.22 (95% CI: 0.80–1.86).

### 3.4. Publication Bias

A funnel plot for the comparison that involved 12 studies (antibiotic-loaded cement vs. plain cemented fixation) showed visual evidence of publication bias (Appendix A), which was consistent with Egger’s regression symmetry test (*p* = 0.038). Using the trim-and-fill method did not impute any artificial studies into the meta-analysis.

## 4. Discussion

### 4.1. Key Findings

Based on a systematic review and meta-analysis of observational and interventional evidence, we have evaluated the body of evidence linking cemented, uncemented, and hybrid fixation methods with the risk of PJI following primary TKR. Pooled evidence from observational studies suggests that uncemented fixation is associated with lower overall PJI risk when compared with cemented fixation. This reduction in risk, however, lost significance when the largest study [14] was excluded. There were no differences in PJI risk when hybrid fixation was compared with cemented or uncemented fixation. There was no significant difference in overall PJI risk when antibiotic-loaded cemented fixation was compared with plain cemented fixation. However, in analysis limited to the first 6 postoperative months of follow-up, antibiotic-loaded cement was associated with an increased PJI risk when compared with plain cemented fixation. Subgroup analyses involving the comparison between antibiotic-loaded cemented and plain cemented fixation showed evidence of effect modification by geographical location and population source. Antibiotic-loaded cement compared with plain cemented fixations was associated with decreased PJI risk in Asian populations. However, given the limited number of studies available for these subgroup analyses, the results need to be interpreted with caution. Finally, limited data from RCTs showed no differences in PJI risk between fixation types.

### 4.2. Comparison with Previous Work

To our knowledge, no previous reviews have evaluated the associations of all fixation types with the risk of PJI following primary TKR; therefore, it is difficult to make a head-to-head comparison in the context of previously published work beyond the papers included in this analysis. However, a number of reviews have compared uncemented versus cemented fixations or antibiotic-loaded cemented versus plain cemented fixations. Two reviews compared cemented fixation with uncemented fixation in terms of implant survival but did not compare infection outcomes between the fixation methods [32,33]. In a pooled analysis of five studies, Wang and colleagues showed no difference between cemented and uncemented fixation with respect to infection [34]. Consistent with our findings, several published meta-analyses of observational studies and RCTs have also not demonstrated any difference in the overall incidence of infection between antibiotic-loaded cement and plain cement fixations in primary TKR [35,36,37]. However, based on a larger number of studies and more detailed analyses, our review presents new observational findings which show that uncemented fixations are associated with lower PJI risk when compared with cemented fixations (albeit on inclusion of the large NJR study [14]) and the effects of antibiotic-loaded and plain cemented fixations seem to depend on the timing of the postoperative period following primary TKR, geographical location, and the source of the data. In our recent review conducted in primary THR patients, we have also shown that uncemented fixations are associated with lower PJI risk when compared with cemented fixations [15]. Consistent with the data in THR patients, the evidence from RCTs in knee patients is also limited and inconclusive.

### 4.3. Possible Explanations for Findings

Compared to uncemented prostheses, cemented prostheses may cause an increased risk of infection via a number of pathways. Evidence from studies conducted in THR patients suggest the bone necrosis caused by direct toxicity or generation of heat during the cemented polymerization process [38] may create conditions conducive for bacterial growth [39,40]; although in TKR, it is unlikely that the cement mantle thickness reaches the threshold required to lead to osteonecrosis. Compared to uncemented TKR, cemented TKR has a longer operating room time [41], which may increase the likelihood of perioperative contamination [42]. One would expect that antibiotic-loaded bone cement should confer a lower risk of infection compared with plain bone cement, due to the elution of antibiotics from the bone cement [43]. However, when the overall evidence was considered, antibiotic-loaded cemented fixations seemed to be associated with increased PJI risk in the early postoperative period but not at longer-term follow-up. The elution of antibiotics may only achieve effective concentrations against certain bacteria or for very short postoperative time periods [43]. Emergence of antimicrobial resistance could be an explanation; in vitro data suggests that prolonged exposure of micro-organisms to subinhibitory concentrations of antibiotics promotes mutations that confer resistance [44,45]. Evidence from both animal and human studies show high rates of antibiotic-resistant infections in antibiotic-loaded cement [46,47]. Given that our findings are limited to the observational studies included, it could be that our observations reflect appropriate selection of low risk patients only to receive plain cement with higher risk patients receiving antibiotic-loaded cement, with this selection occurring on the basis of factors not included in, or adjusted for, in the studies included. Other possible explanations include biases in study designs, such as misdiagnosis of PJI and inability to account for important risk factors, such as age, sex, and comorbidities, including diabetes mellitus, nature of prostheses, surgical environment, nature of prior surgical procedures, and other patient factors.

### 4.4. Implications of our Findings

Though prosthesis design and materials are constantly evolving, the ideal fixation method for TKR is still under considerable debate [41], and this is because of surgical preferences and inconsistencies in clinical outcomes reported. Cemented fixation in TKR has been regarded as the gold standard for several decades, given the extensive evidence on its good clinical outcomes. Discouraging initial results for uncemented TKRs [48,49] led to a decline in use. However, with the development of new materials and prosthetic designs, the use of uncemented fixation is becoming an attractive option among surgeons [50]. Emerging evidence suggests that modern uncemented knee prostheses have comparable survivorship and clinical outcomes to cemented prostheses. In addition, cemented fixations have drawbacks, which include longer surgical time, possibility of thermal osteonecrosis, and complex revisions in the event of a failure [51]. Uncemented fixation, commonly used in younger patients and those with good bone quality, is approximately three times more expensive than cemented fixation; however, they have many advantages, which include shorter operative time, providing a biologic interface between bone and implant leading to a durable fixation, preservation of bone stock, reduced risk of cement-related complications, such as third body wear from retained loose fragments, ease of revision in the event of a failure [41,51], and in addition, a lower risk of PJI. There is a changing demography in the TKR population, as the population with osteoarthritis is getting younger [52]. Given the large projected increases in the numbers of TKRs that will be performed [11], the incidence of PJI is also expected to rise. It appears cemented fixation may be associated with an increased risk of PJI when compared to uncemented fixation. If equivalent outcomes in terms of revision for other indications can be achieved with modern uncemented TKR when compared to cemented TKR, then it may be reasonable to recommend surgeons to use an expensive uncemented fixation (especially for patients at high risk of revision), as it gives comparable clinical outcomes to cemented fixation, has several advantages, and is also associated with lower PJI risk. However, the current evidence suggests a higher overall rate of revision for uncemented TKR compared to cemented TKR [1]. The efficacy of antibiotic-loaded cement compared with plain cement in decreasing infection has been demonstrated in studies of primary THR [15]. Apart from reasons such as the development of antibiotic resistance, biases of the study designs, and chance findings, it is difficult to explain the contrasting findings we have demonstrated in the early postoperative period for TKRs. However, there is a possibility that antibiotic-loaded cement fixation has no effect on the prevention of PJI after primary TKA. Wang and colleagues, in their combined analysis of 2293 patients and critical review of seven articles, concluded that antibiotic-loaded bone cement had no effect on PJI prevention compared with plain cement in primary TKR [53]. The authors recommended that since previous studies were based on short-term follow-up (12 months), further studies with long-term follow-up were needed. Even though the evidence base is limited, antibiotic-loaded cement is commonly used for primary TKR throughout Europe, whereas in the United States it is mostly used for treating revision for infection in TKR. Recommendations against its use for primary joint surgery in North America are based on concerns regarding high costs, allergic reactions, toxicity, and antibiotic resistance [29,54,55,56]. We acknowledge that much of the evidence is based on observational data, which are limited by biases such as selection bias and reporting, residual confounding, and uncertainty with coding of fixation types; hence, the results should be interpreted with caution. Definitive RCTs with long-term follow-ups are warranted to confirm or refute these findings.

### 4.5. Study Strengths and Limitations

We have conducted a comprehensive systematic review and meta-analysis that evaluates the relationships of cemented, uncemented, and hybrid fixation methods with PJI risk following primary TKR. We employed a comprehensive search strategy across multiple databases, as well as manual searches of relevant articles, thereby identifying several additional observational and intervention studies conducted on the topic. We were able to harmonize the data to a consistent comparison to enable pooling, and this enhanced interpretation of our findings. In addition, we extracted detailed data that enabled reporting of estimates for specific time periods, exploration of heterogeneity, and the assessment of effect modification where possible. Finally, a detailed assessment of the quality of the included studies (including risk of bias) was conducted using established and validated tools. There were several limitations to the current study, the majority of which were related to the studies included in the review, and these should be taken into account when interpreting the results. The lack of reporting or the heterogeneous definition of PJI employed by included studies could have limited the validity of the findings. In registry studies, it is well known that PJI diagnosis reflects the clinical judgement of the surgeon and there are issues relating to under-reporting of revision for PJI, thus yielding potentially lower incidence estimates of PJI [57]. There were a limited number of studies (<10) for most of the comparisons reported and these precluded assessment of heterogeneity and effect modification. The large registry study [14] contributed to the beneficial effect seen in uncemented fixations. Most of the evidence comes from observational study designs, which are unable to directly prove causation. In addition, the majority of reported risk estimates were confounded as they were estimated from the raw data. Finally, some of the studies were conducted decades ago and inclusion of these data do not take into account the evolving nature of prosthetic materials, surgical procedures, as well as contemporary antibiotic prophylaxis.

## 5. Conclusions

Aggregate observational evidence suggests uncemented primary TKR may be associated with lower PJI risk compared with cemented primary TKR. In the early postoperative period, antibiotic-loaded cemented fixation may be associated with increased PJI risk when compared with plain cement. There are no differences in PJI risk when hybrid fixations are compared with cemented or uncemented fixations. Data from RCTs is limited and uncertain.

## Figures and Tables

**Figure 1 jcm-08-00828-f001:**
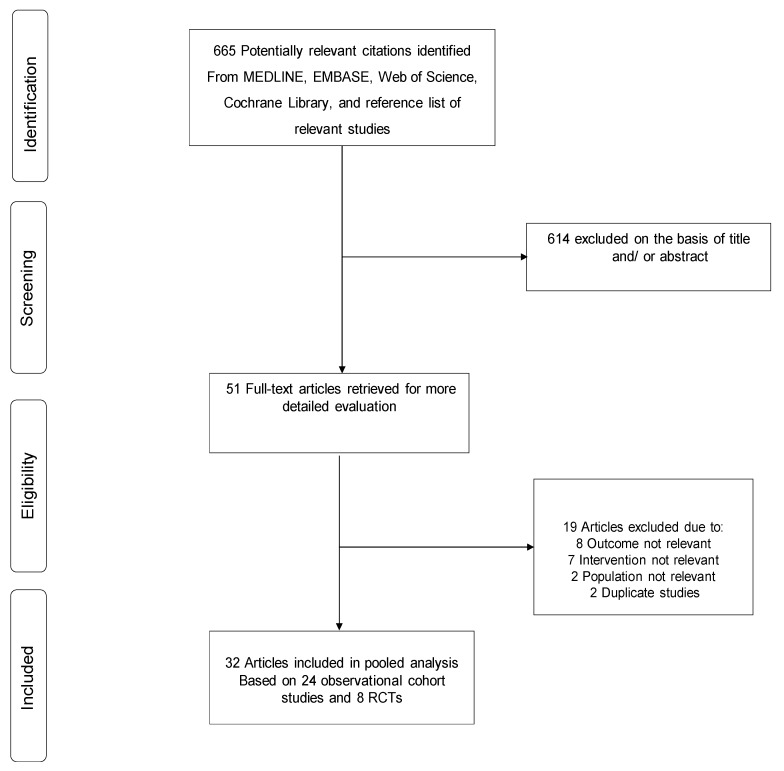
PRISMA flow diagram; Note: RCT, randomised controlled trial.

**Figure 2 jcm-08-00828-f002:**
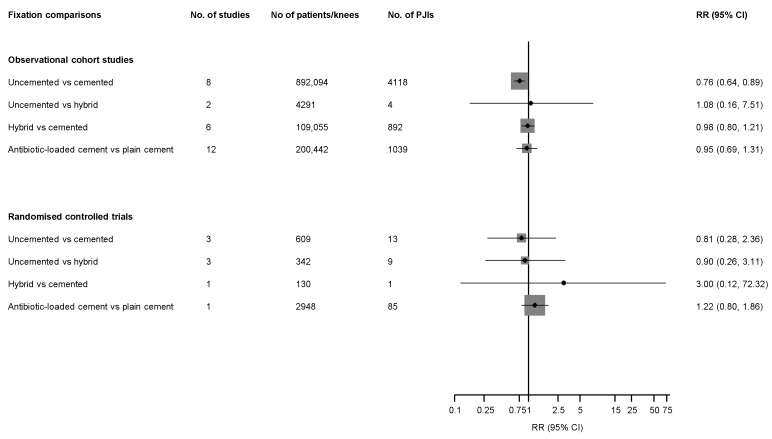
Fixation types in primary total knee replacement and risk of prosthetic joint infection in observational studies and randomised controlled trials. Note: CI, confidence interval (bars); PJI, prosthetic joint infection; RR, relative risk.

**Figure 3 jcm-08-00828-f003:**
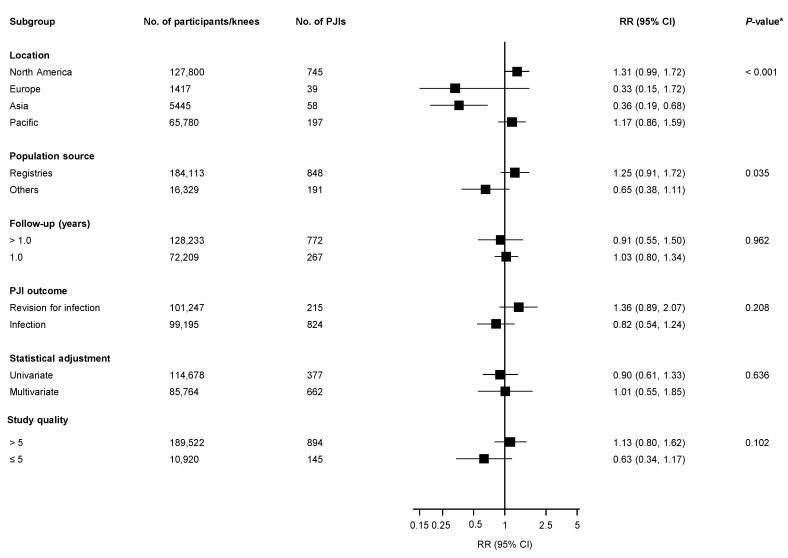
Comparison of all antibiotic-loaded cemented fixation with plain cemented fixation in primary total knee replacement and the risk of prosthetic joint infection in observational studies, grouped according to several study characteristics. Note: CI, confidence interval (bars); PJI, prosthetic joint infection; RR, relative risk; *, *p*-value for meta-regression.

**Table 1 jcm-08-00828-t001:** Characteristics of studies included in review.

Author, Year of Publication	Year of Study	Country	Indication for Total Hip Replacement	Average Age (Years)	Design, Source of Data	Fixation Types Compared	Mean/Median Follow-Up Duration, Years	No. of Participants/Knees	Infection Outcome Reported (Definition)	No. of PJIs	Study Quality
Wilson, 1990	1973–1987	U.S.	NR	NR	Observational cohort, Hospital	Uncemented, cemented, hybrid	Up to 6.0 years	4171	Deep infection (Purulent material obtained from joint and positive bacterial culture)	67	4
Duffy, 1998	1985–1987	U.S.	Uncemented (OA 76.4%; RA 16.4%; PTA 5.5%; ankylosing spondylitis 1.8%); Cemented (OA 82.4%; RA 11.8%; ancient sepsis 3.9%; osteonecrosis 2.0%)	59.6	Retrospective cohort	Uncemented, cemented	10.0	106	Revision for infection (NR)	1	4
McCaskie, 1998	1987–1990	U.K.	Cemented (OA 84%); Uncemented (OA 86%)	68.8–70.2	RCT, Hospital	Uncemented, cemented	5.0	113	Infection (NR)	1	NA
Pecina, 2000	1985–1991	Croatia	OA 68.3%; RA 31.7%	61.0	Observational cohort	Uncemented, cemented, hybrid	7.3	142	Revision for infection (NR)	5	5
Eveillard, 2003	1995–1999	France	NR	NR	Observational cohort, Hospital	Antibiotic loaded cement, plain cement	At least 1 year	167	Infection (Isolation of organisms from tissue sample; confirmed by surgeon)	9	5
Baker, 2007	1987–1997	U.K.	OA 91.4%; RA 7.6%; other 1.0%	70.5	RCT	Uncemented, cemented	8.7–8.9	396	Revision for infection (NR)	11	NA
Beaupre, 2007	1996–2000	Canada	Non-inflammatory arthritis 100%	63.4	RCT	Uncemented, hybrid	5.0	81	Infection (NR)	6	NA
Jamsen, 2009	1997–2004	Finland	Primary OA 87.9%; secondary OA 2.7%; RA 7.6%; other arthritis 1.0%; other 0.8%	71.0	Retrospective cohort, FAR and FHDR	Uncemented, cemented, hybrid	3.1	40,135	Revision for infection (NR)	387	7
Dowsey, 2009	1998–2005	Australia	OA 91.8%; RA 7.8%; osteonecrosis 0.2%; trauma 0.2%	72.0	Retrospective cohort, Institutional database	Antibiotic loaded cement, plain cement	1.0	1214	PJI (CDC criteria)	18	6
Ghandi, 2009	1998–2006	Canada	Primary or secondary OA; RA	66.1	Retrospective cohort, Hospital	Antibiotic loaded cement, plain cement	1.0	1625	Deep infection (CDC criteria)	43	5
Namba, 2009	2003–2007	U.S.	OA 92.4%; other 7.6%	68.0	Retrospective cohort, community-based registry	Antibiotic loaded cement, plain cement	NR	22,889	Deep infection (CDC criteria)	182	8
Demey, 2011	2004–2005	France	OA (96.9%); chondrocalcinosis (3.1%)	72.3	RCT	Hybrid, cemented	2.7–2.8	130	Deep infection (NR)	1	NA
Namba, 2013	2001–2009	U.S.	OA 96.8%; PTA 1.2%; RA 2.2%; osteonecrosis 0.4%; other 0.9%	67.4	Retrospective cohort, Registry	Antibiotic loaded cement, plain cement	NR	56,216	Deep SSI (CDC criteria)	404	8
Lass, 2013	2003–2007	Austria	Idiopathic arthritis 88.3%; PTA 5.0%; RA 3.3%; avascular necrosis 0.8%	66.9	Observational cohort	Uncemented, hybrid	5.0	120	Revision for infection (NR)	1	5
Pelt, 2013	NR	U.S.	Hybrid (OA 95%; RA 2%; PTA 3%; other 0%); Cemented (OA 90%; RA 7%; PTA 2%; other 1%)	59.3–65.9	Observational cohort	Hybrid, cemented	3.2–4.1	304	Revision for sepsis (NR)	5	5
Hinarejos, 2013	2005–2010	Spain	NR	75.9	RCT	Antibiotic loaded cement, plain cement	3.2	2948	Deep and superficial infection (CDC criteria)	85	NA
Qadir, 2014	2000–2010	U.S.	NR	68.1	Retrospective cohort, Institutional registry	Antibiotic loaded cement, plain cement	1.0	2511	Infection (CDC criteria)	17	6
Gutowski, 2014	2000–2002; 2004–2007	U.S.	NR	65.8	Retrospective cohort, Hospital	Antibiotic loaded cement, plain cement	Over a 3.0-year period	7878	PJI (MSIS criteria)	63	5
Bohm, 2014	2003–2008	Canada	OA 100%	70.0	Retrospective cohort, CIHI and CJRR	Antibiotic loaded cement, plain cement	2.0	36,681	Revision for infection (NR)	36	6
Choy, 2014	2002–2004	Korea	OA 100%	67.8	RCT	Uncemented, hybrid	9.5	168	SSI (NR)	2	NA
Lizaur-Utrilla, 2014	1999–2007	Spain	OA (92.5%); PTA (7.5%)	51.7	RCT	Uncemented, hybrid	7.1	93	Deep wound infection (NR)	1	NA
Petursson, 2015	1999–2012	Norway	Primary OA 90%; other 10%	69.0	Observational cohort, NAR	Hybrid, cemented	11.0	24,680	Revision for infection (NR)	217	7
Wang, 2015	2003–2012	China	OA 87.8%; other 12.2%	64.8	Retrospective cohort, Hospital	Antibiotic loaded cement, plain cement	1.0	2293	Deep infection (CDC criteria)	10	6
Fricka, 2015	2010–2012	U.S.	NR	58.6–60.2	RCT	Uncemented, cemented	2.0	100	PJI (NR)	1	NA
Tayton, 2016	1999–2012	New Zealand	OA 95%; AVN 0.3%; Trauma 1.2%; RA 3.4%; other 0.2%	<55 to >75 *	Prospective cohort, New Zealand Joint Registry	Antibiotic-loaded cement, plain cement	1.0	64,566	Revision for infection (NR)	179	7
Wu, 2016	2009–2013	Taiwan	OA, RA, PTA	69.7	Retrospective cohort	Antibiotic-loaded cement, plain cement	1.0–5.0	3152	SSI (CDC criteria)	48	6
Prudhon, 2017	2003–2006	France	OA 88.5%; post-traumatic OA 3.0%; RA 4.5%; patellofemoral OA 4.0%	73.0	Observational cohort	Uncemented, cemented	12.1–13.7	200	Infection (NR)	1	5
Sanz-Ruiz, 2017	2009–2012	Spain	NR	76.1–76.4	Prospective cohort	Antibiotic-loaded cement, plain cement	2.0 (minimum)	1250	Infection (MSIS criteria)	30	4
Vertullo, 2018	1999–2015	Australia	OA 100%	69.0	Observational cohort, Registry	Hybrid, cemented	13.0	39,623	Revision for infection (NR)	215	7
Gwam, 2018	2015	U.S.	OA 100%	65.8	Retrospective cohort, NIS database	Uncemented, cemented	NR	167,930	SSI (NR)	NR	5
Lenguerrand, 2018	2003–2013	U.K.	OA (97.3%); other (2.7%)	69.0	Prospective cohort, Registry	Uncemented, cemented	4.6	679,010	Revision for infection (NR)	3227	7
Miller, 2018	2013–2014	U.S.	NR	64.4	Institutional database	Uncemented, cemented	2.4–5.3	400	Infection (NR)	1	5

Note: *, age range of participants; CDC, Centres for Disease Control Prevention; CIHI, Canadian Institute for Health Information; CJRR, Canadian Joint Replacement Registry; FAR, Finnish Arthroplasty Register; FHDR, Finnish Hospital Discharge Register; MSIS, Musculoskeletal Infection Society; NA, not applicable; NAR, Norwegian Arthroplasty Register; NIS, National Inpatient Sample; NR, not reported; OA, osteoarthritis; PJI, prosthetic joint infection; PTA, post-traumatic arthritis; RA, rheumatoid arthritis; RCT, randomised controlled trial; SSI, surgical site infection.

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
