# Peer review of "Influence of Fixation Methods on Prosthetic Joint Infection Following Primary Total Knee Replacement: Meta-Analysis of Observational Cohort and Randomised Intervention Studies"

_jcm, 2019, doi:10.3390/jcm8060828_

Round 1
Reviewer 1 Report
This is an outstanding systematic review and meta-analysis assessing published evidence linking type of fixation (cemented, uncemented, or hybrid) with the risk of PJI following primary TKR, which identified an increased risk of infection in cemented versus uncemented fixation. This finding is important and timely, as the use of antibiotic cement was seriously challenged at the 2018 International Consensus Meeting on Musculoskeletal Infection (ICM-MI), based on lack of published efficacy and our current understanding of biofilm bacteria (Saeed K, et al. 2019. 2018 international consensus meeting on musculoskeletal infection: Summary from the biofilm workgroup and consensus on biofilm related musculoskeletal infections. J Orthop Res 37:1007-1017).
The only concern is that the authors cite a 15 year old reference and a national registry to substantiate the 0.5-4% infection rates of primary TKRs. However, the current incidence of TKA PJI of 0.4-1.5% was established by the 2018 International Consensus Meeting on Musculoskeletal Infection (Schwarz EM, et al. 2019. 2018 International Consensus Meeting on Musculoskeletal Infection: Research Priorities from the General Assembly Questions. J Orthop Res 37:997-1006). This should be corrected.
Author Response
This is an outstanding systematic review and meta-analysis assessing published evidence linking type of fixation (cemented, uncemented, or hybrid) with the risk of PJI following primary TKR, which identified an increased risk of infection in cemented versus uncemented fixation. This finding is important and timely, as the use of antibiotic cement was seriously challenged at the 2018 International Consensus Meeting on Musculoskeletal Infection (ICM-MI), based on lack of published efficacy and our current understanding of biofilm bacteria (Saeed K, et al. 2019. 2018 international consensus meeting on musculoskeletal infection: Summary from the biofilm workgroup and consensus on biofilm related musculoskeletal infections. J Orthop Res 37:1007-1017).
REPLY. We thank the Reviewer for the generous comments regarding our article. We agree that these findings are timely given the recent controversy at the 2018 International Consensus Meeting on Musculoskeletal Infection (ICM-MI).
The only concern is that the authors cite a 15 year old reference and a national registry to substantiate the 0.5-4% infection rates of primary TKRs. However, the current incidence of TKA PJI of 0.4-1.5% was established by the 2018 International Consensus Meeting on Musculoskeletal Infection (Schwarz EM, et al. 2019. 2018 International Consensus Meeting on Musculoskeletal Infection: Research Priorities from the General Assembly Questions. J Orthop Res 37:997-1006). This should be corrected.
REPLY. Thank you for this important comment. This has now been corrected.
Reviewer 2 Report
The authors present a meticulously performed meta- analysis regarding a relevant clinical question.
The material and method section is especially well crafted.
In my opinion the author should discuss differences in the definition of outcome (PJI/SSI/Deep Infection/ Revision surgery) more detailled in the limitations section and the sensitivity of identification of PJIs in registries.
In addition to the outcome at 6month follow up, I would be interested in the outcome with a follow up of 24month (which many of the studies included provide). This would include the majority of infections that were caused by contamination during or directly after Implantation.
Author Response
The authors present a meticulously performed meta- analysis regarding a relevant clinical question. The material and method section is especially well crafted.
REPLY. We thank the Reviewer for the compliments. It is very much appreciated.
In my opinion the author should discuss differences in the definition of outcome (PJI/SSI/Deep Infection/ Revision surgery) more detailed in the limitations section and the sensitivity of identification of PJIs in registries. REPLY. As suggested, we have included a discussion on the limitations related to the diagnosis and definition of PJI. We have updated the Results section and Table 1 on studies that provided a definition for the outcome.
In addition to the outcome at 6month follow up, I would be interested in the outcome with a follow up of 24month (which many of the studies included provide). This would include the majority of infections that were caused by contamination during or directly after Implantation.
REPLY. Thank you for this comment. As suggested, we have now presented results at 24 months for the comparison between antibiotic-loaded cemented fixation and plain cemented fixation. No significant difference in PJI was demonstrated.